# Put a Bow on It: Knotted Antibiotics Take Center Stage

**DOI:** 10.3390/antibiotics8030117

**Published:** 2019-08-11

**Authors:** Stephanie Tan, Gaelen Moore, Justin Nodwell

**Affiliations:** Department of Biochemistry, MaRS Discovery District, University of Toronto, 661 University Avenue, Toronto, ON M5G 1M1, Canada

**Keywords:** lasso peptides, ribosomally synthesized post translationally modified peptide (RiPP), antibiotic, target, mechanism of action

## Abstract

Ribosomally-synthesized and post-translationally modified peptides (RiPPs) are a large class of natural products produced across all domains of life. The lasso peptides, a subclass of RiPPs with a lasso-like structure, are structurally and functionally unique compared to other known peptide antibiotics in that the linear peptide is literally “tied in a knot” during its post-translational maturation. This underexplored class of peptides brings chemical diversity and unique modes of action to the antibiotic space. To date, eight different lasso peptides have been shown to target three known molecular machines: RNA polymerase, the lipid II precursor in peptidoglycan biosynthesis, and the ClpC_1_ subunit of the Clp protease involved in protein homeostasis. Here, we discuss the current knowledge on lasso peptide biosynthesis as well as their antibiotic activity, molecular targets, and mechanisms of action.

## 1. Introduction

The natural products known as secondary or specialized metabolites are the source of thousands of antibiotics and other drugs that are critical for current medical practice. The majority of these compounds originated from the *Actinobacteria*, especially the *streptomycetes*, as well as lower fungi. The numbers of compounds produced by each species ranges from less than five in most cases to as many as 50 in the *Actinobacteria* and fungi. These compounds have a broad range of biological activities including many with therapeutic potential. However, many of the secondary metabolites encoded in the genome database remain unknown as their structures and biological activities have not yet been identified. Gaining access to this chemical space will provide us more resources to combat antimicrobial resistance and potential other areas of unmet medical need [1,2,3,4]. In addition, characterizing these molecules will shed light on poorly understood mechanisms and enzymes, a valuable goal on the road to new treatments for bacterial pathogens.

Over 60% of the antimicrobials brought to market are natural products or derivatives thereof [5]. Peptide antibiotics are a subset of natural products and can be of ribosomal or non-ribosomal origin. While non-ribosomal peptide antibiotics such as the β-lactams and the glycopeptides are familiar, the ribosomally-synthesized and post-translationally modified peptides (RiPPs) are less well understood [6]. The RiPPs are a large class of chemically and functionally diverse peptides which includes members such as nisin, a food preservative, thiostrepton, an antibacterial used in veterinary medicine, and omega-conotoxin MVIIA, which has been developed into a synthetic analgesic drug [7,8,9,10]. The RiPP family contains impressive structural diversity, introduced by post-translational modifications, which gives this class of peptides a spectrum of activities ranging from antibacterial to morphogenic to analgesic.

To our knowledge, none of the known RiPPs have found clinical application as antimicrobials at this time; relatively few have found any technical application at all outside the laboratory. The lanthipeptide nisin is an exception: it is used as a food additive [11]. One possible road block to serious consideration of the RiPPs is that, because of their large size (usually > 1.5 kDa), numerous H-bond acceptor and donor groups, and other properties, virtually none of them obey Lipinski’s rule of five [12]. We argue that this departure from convention is part of what is so attractive about these molecules: they fall well outside of the known antibiotic chemical space. Indeed, the very successful antibiotics vancomycin and daptomycin also disobey the rule of five and there have been many arguments that these rules are too restrictive for antibiotic discovery [13]. As we will show, while many RiPPs interact with well-known antibiotic targets, others target novel pathways. At the very least therefore, these molecules are powerful chemical probes pointing to new biology that can be targeted for antibacterial therapy. We would go one step further by predicting that there will eventually be RiPPs or RiPP derivatives in clinical use.

While the RiPPs are structurally and functionally diverse, the family contains a unifying biosynthetic logic. All RiPPs are genomically encoded as a precursor peptide which consists of an N-terminal (or occasionally C-terminal) leader sequence and a C-terminal core sequence (Figure 1). The leader sequence enables the recognition of the RiPP precursor by the enzymes that perform post-translational modifications in the C-terminal core. The mature RiPP is produced when the post-translational modifications of the C-terminal sequence are complete and the N-terminal sequence is cleaved off. Some RiPP gene clusters encode dedicated transport and regulation genes [6]. Using this strategy, the RiPPs natural products achieve impressive chemical and functional diversity. While others have reviewed the various sub-families of RiPPs [14,15,16], in this review we will focus on one of the newest sub-families of RiPPs, the lasso peptides.

## 2. Lasso Peptides

The lasso peptides are a large family of chemically diverse and poorly understood RiPPs. The first member of this family, anantin, was identified less than 30 years ago [17]. Lasso peptides range from 14 to 24 amino acid residues in length and contain a characteristic isopeptide bond between the N-terminal amine and the carboxy group of a glutamate or aspartate residue in the 7th, 8th, or 9th position (Figure 2B)-this creates an N-terminal macrocycle [18]. The C-terminal tail is threaded through the N-terminal macrocycle, producing a topology which resembles a lasso. In some lasso peptides, large, bulky amino acids act as “steric plugs” to prevent unthreading of the C-terminal tail from the N-terminal macrocycle. This lasso topology provides stability and as a result, the lasso peptides are resistant to proteolysis and chemical degradation. Lasso peptides are also intrinsically resistant to temperatures as high as 95–120 °C [19,20], however some are heat-labile like astexin-1 and begin to unthread at 50 °C [21]. Their heat-stability was initially thought to be a feature of the lariat-knot topology. Instead, thermic stability of the lasso peptide depends on the general flexibility of the C-terminal tail, the size of the macrolactam ring and the size and position of the “steric plug” amino acids [22,23].

The lasso peptides are further subdivided into four classes based on the occurrence of intramolecular disulfide bridges. Class I lasso peptides contain two disulfide bridges linking the macrocyclic ring with the threaded tail, whereas the class II lasso peptides have no disulfide bridges. The class III and class IV lasso peptides each contain one disulfide bridge at different positions within their structure. In class III peptides, the disulfide links the N-terminal ring and the C-terminal tail, while in class IV peptides, the disulfide links the C-terminal tail to itself (Figure 2C). As of 2017, 96% of lasso peptides were predicted to belong to class II based on all genome sequences available at the time [24]. To date, the vast majority of lasso peptides that have been investigated belong to class II and the classes III and IV include only one member each [25]. Many of the lasso peptides, particularly those belonging to class II, have been studied by overexpression in a heterologous host organism, typically the proteobacterium *Escherichia coli*. It is not understood why class II lasso peptides are more effectively produced in an *E. coli* heterologous system, however those that have are also of proteobacterial origin [20,21,22,26]. The regulation of lasso peptide production strains remains poorly understood but has been reviewed to date by Li et al. [27].

Lasso peptides, like other classes of RiPPs, are produced across many phyla of bacteria, particularly by Actinobacteria, Proteobacteria, and Firmicutes. To take one example, the genus *Streptomyces* is known as a prolific producer of polyketides and nonribosomal peptides, with most species having genes to produce 20–40 such compounds. In contrast, this genus is not generally known as a producer of RiPPs. To investigate the ability of *Streptomyces* to produce RiPP natural products, we randomly selected strains of *Streptomyces* and searched their sequenced genomes for RiPP biosynthetic gene clusters using the program AntiSMASH 5.0 [28]. In addition to lasso peptides, we chose to search for lanthipeptides, a class of RiPPs containing characteristic lanthionine and methyllanthionine amino acids, and thiopeptides, a class of RiPPs with sulfur-rich heterocycles and a unique, central nitrogen-containing heterocycle. We found that 19 of the 50 genomes encoded at least one lasso peptide cluster, with some genomes encoding as many as three separate lasso peptide clusters (Figure 3). Lanthipeptide clusters were the most abundant in the 50 sampled genomes; nearly all genomes contained at least one lanthipeptide cluster and several strains contained four lanthipeptide clusters. Thiopeptide clusters were approximately equal in abundance to the lasso peptide clusters. Several strains were found to contain other RiPP clusters, which are not depicted on Figure 3 for the sake of simplicity. The occurrence of these RiPP clusters is displayed on a cladogram constructed using FastTree 2.1 and visualized using the interactive Tree of Life [29,30]. Remarkably, the streptomycetes, which have been mined for natural products since the 1930s appear to also be prolific producers of RiPPs, with some strains in our analysis encoding as many as eight distinct RiPPs. We surmise that this is likely to be the case for many other genera of bacteria in nature.

### Lasso Peptide Biosynthesis

Lasso peptide biosynthesis requires at least two enzymes: a leader peptidase and a lasso cyclase [31]. Analogous to other RiPP clusters, lasso peptide clusters encode a cysteine peptidase that serves to cut the leader sequence from the precursor peptide (Figure 2A). The proteolysis step in some cases requires ATP though this is likely a requirement for lasso peptide pre-folding, not for proteolysis [32]. The next step in lasso peptide biosynthesis is cyclisation, which is catalyzed by an ATP-dependent lasso cyclase. The lasso cyclase is predicted to catalyze the adenylation of the carboxylic acid of the glutamate or aspartate and the free N-terminal amine acts as a nucleophile to attack the adenylated carboxylic acid, releasing AMP (Figure 2B).

The lasso cyclases are homologous to AsnB, one of two *E. coli* asparagine synthetases, which uses aspartate as a substrate to produce asparagine. About 70% of lasso cyclases cyclize aspartate-containing precursor peptides while the remaining 30% cyclize glutamate-containing precursor peptides. Importantly, all lasso peptide clusters have a RiPP precursor peptide recognition element (RRE) present, which is important for leader peptide recognition within the precursor peptide and recruitment of the peptidase to the precursor peptide for cleavage (Figure 4). RREs, which possess a winged helix-turn-helix structure, are broadly distributed across RiPP gene clusters and are important for biosynthesis across many classes of RiPPs [33]. In lasso peptide clusters the RRE, also known as the B1 protein is sometimes (but not always) fused to the N-terminal region of the leader peptidase. In cases where the RRE and leader peptidase are separately encoded, the RRE interacts with the leader peptidase to guide the peptidase to its target cleavage site. It has been suggested that this RRE-peptidase interaction is held together by electrostatic interactions [34]. Lasso peptide gene clusters often (though not always) encode ABC transporters. Additional post-translation modifications such as methylation, phosphorylation, and acetylation have also been reported [35,36,37].

Lasso peptide precursors have several conserved features that are important for their biosynthesis and bioactivity. A conserved YxxP motif and threonine residue in the leader peptide sequence are required for leader peptide recognition by the RRE [24]. Recently, the crystal structure of the leader portion of the precursor peptide of fusilassin/fuscanodin, bound to its cognate RRE, TfuB1, was determined [44]. This resolved the outstanding question of how the RRE and leader peptidase interact with the precursor peptide to produce the mature peptide. Sumida et al. showed that the YxxP motif, as well as a highly conserved Leu, fit tightly into a hydrophobic cleft formed by the RRE protein. This interaction brings the precursor peptide and TfuB1 together to form a hydrophobic patch found to be necessary for cleavage activity by the lasso peptidase. The hydrophobic patch includes two key residues, Phe-6 in the leader peptide and Tyr-33 in TfuB1. This is highly conserved amongst diverse producers of lasso peptides including the thermophilic species of Actinobacteria, *Thermobifida fusca*, and the Firmicute, *Bacillus pseudomycoides* [44].

It would be expected for other leader peptides and their respective RRE to interact similarly as the residues involved in forming the hydrophobic patch are highly conserved. Interestingly, a sequence alignment of the antibacterial lasso peptides produced by Actinobacteria have the conserved YxxPxL motif in their leader peptide sequence. This conserved sequence is found in all other lasso peptides produced by Actinobacteria, except for albusnodin [24,37,45,46]. By contrast, Proteobacteria encode a leader peptide with an I/V/LxxxA motif upstream of the cleavage site (Figure 4A). The interacting Phe-6 in fusilassin’s leader peptide is also highly conserved which suggests this amino acid has co-evolved alongside its interacting partner, B1, at Tyr-33 in Actinobacteria. A threonine is observed two amino acids upstream of the cleavage site in all lasso peptides, as expected [24]. A similar analysis was done with the RRE domain. The RRE domain is sometimes fused to the N-terminal of the leader peptidase. In the case of Actinobacteria, all clusters encoding a lasso peptide have the RRE domain encoded separately from the leader peptidase. Highly conserved residues are found in all the RREs that have been implicated in interacting with the leader peptide [44] (Figure 4B).

In contrast, Proteobacteria have only one gene encoding both the RRE and the peptidase domain. Recognition of the precursor peptide by the B protein has not been investigated in detail; cleavage of the leader peptide by the C-terminal peptidase has been thoroughly characterized [32]. Alignment of five different lasso peptides suggest two conserved portions may be important in recognizing the leader peptide. The first involves an important aspartic acid found in between a patch of isoleucines, leucines, and valines. The other conserved part includes an Arg-Trp, except for capistruin which has Arg-Gly-Trp (Figure 4B). Structural characterization of the N-terminal region of the B protein in complex with the leader peptide will be able to differentiate how the Proteobacteria lasso peptides process the molecule from Actinobacteria.

In the core peptide, the first position of the sequence is often a glycine, although many other initial amino acids have been reported such as cysteine, serine, alanine, and tryptophan [24,47] (Figure 4A). The carboxylate macrocycle acceptor is often located in the 8th or 9th position of the core sequence, although the 7th position is also possible [34].

Genes encoding putative lasso peptides are widely distributed in prokaryotes, including Proteobacteria, Actinobacteria, Firmicutes, and Archaea [24]. To date, only roughly 50 have been studied, underscoring that this is an underexplored region of chemical space. The discovery of novel lasso peptides by mining genomic data has proven to be especially fruitful—several tools have been developed for this purpose [24,28,48,49,50]. Structurally diverse lasso peptides have been reported in streptomycetes including siamycin, anantin, streptomonomicin, and BI-32169.

## 3. Targets of Lasso Peptides

Diverse types of bioactivity have been observed amongst the known lasso peptides, including many that are medically relevant. Most importantly, antimicrobial activity against both Gram-positive and Gram-negative pathogens have been observed for at least 14 of these compounds. Other targets include the endothelin B receptor, relevant to numerous cancers like breast, stomach, colon, and prostate [51] and the glucagon receptor associated with diabetes [25]. In addition, cancer cell invasion in human lung cells can be inhibited by a lasso peptide [52].

Among these antibacterials, the bioactivity ranges significantly as some lasso peptides such as lassomycin and lariatin A/B exhibit bioactivity exclusively against *Mycobacterium* sp., whereas others such as siamycin-I and LP2006 act on Gram-positive organisms. The current state of knowledge is summarized in Table 1 [19,24,35,39,40,41,42,43,53,54,55,56,57].

Importantly, only eight antibacterial lasso peptides have been connected to a cellular or molecular target; in most cases, the mechanisms of action remain obscure. Of the compounds that have been studied to date, the three main targets are RNA polymerase, ClpC_1_ of the ClpC_1_P_1_P_2_ protease complex and lipid II, the key precursor molecule of peptidoglycan [35,58,59] (Figure 5).

### 3.1. Inhibitors of Peptidoglycan Biosynthesis

A major antibiotic target in bacteria is its cell envelope including in particular the peptidoglycan. Peptidoglycan is a highly cross-linked polymer that is essential for maintaining cell shape, cell size, and regulating osmotic pressure [60]. So far, two lasso peptides are known to target peptidoglycan biosynthesis: siamycin-I and streptomonomicin.

#### 3.1.1. Siamycin-I

Siamycin-I is a 21-residue, tricyclic peptide produced by the *Streptomyces* genus. The structure of siamycin-I, as determined by NMR, consists of a polar face and non-polar face, giving it an amphipathic character [61]. Its antimicrobial activity was assessed in detail by Daniel-Ivad et al. [39] which showed that a wide range of Gram-positive organisms were sensitive to siamycin-I between 0.9–3.7 µM, including clinically relevant vancomycin-resistant *Enterococci* (VRE) and methicillin-resistant *Staphylococcus aureus* (MRSA) strains at 7.4 µM. It was shown to activate the *lia* (lipid II-interfering antibiotics) operon, suggesting that it might impair peptidoglycan biosynthesis. A detailed analysis into siamycin-I’s target and mechanism of action was further pursued where it was found to bind and inhibit the action of lipid II, the essential precursor molecule involved in peptidoglycan formation [59].

Mutations that confer siamycin-I resistance in *S. aureus* fall primarily in the genes encoding the two-component system (TCS), WalKR, which plays a role in cell wall homeostasis, the recycling of cell wall material and most importantly, the control of cell wall hydrolytic activity [62,63]. Such mutants also exhibit cross-resistance to other lipid-II targeting antibiotics and had a thickened cell wall. Interestingly, mutations in *walKR* are frequently found in vancomycin-intermediate *S. aureus* (VISA) isolates which also exhibit a thickened peptidoglycan layer [64]. Further work on *wal* mutations and their association with antibiotic sensitivity would help illuminate the regulation of cell wall biosynthesis and might provide new routes for overcoming vancomycin resistance in clinical isolates. Enzymatic in vitro studies demonstrated that siamycin-I blocked access to lipid II from the transglycosylation domain of penicillin-binding protein 2 (PBP2), thereby preventing any further incorporation of N-acetylglucosamine (GlcNAc) into the growing glycan [59] (Figure 6).

The action of siamycin-I is unique. Most lipid II inhibitors cause an accumulation of the cytoplasmic precursor, UDP-MurNAc-pentapeptide inside the cytoplasm: siamycin-I is an exception and does not have this effect. Despite its similar target of lipid II, siamycin-I only localizes to the dividing septum of *Staphylococcus aureus* and *Bacillus subtilis* cells, unlike other antibiotics known to target lipid II [65]. These distinct properties underscore the experimental value of these molecules.

Given the limited number of class I lasso peptides and their one or two amino acid differences (Ile, Val, or Leu changes), it might be expected that other class I lasso peptides also inhibit peptidoglycan biosynthesis via lipid II interference. Future studies of siamycin-II, aborycin and RP-71955 would provide more evidence that class I lasso peptides are a novel class of lipid II inhibitors: refining such information with detailed structural studies could shed light on new ways to target this important structure. Mutation of the core peptide sequence would also allow for deeper understanding of residue-specific interactions and what is required for inhibition.

Interestingly, siamycin-I has been also reported to block the access of HIV to its host cell receptors CD4, CCR5, and CXCR4 [66]. Other work has shown blocking of the *fsr* quorum-sensing system in *E. faecalis* and inhibition of ATP-dependent enzymes such as sensor kinases, protein kinases, and ATPases [67,68]. The specificity and biological (or medical) relevance of any of these observations remains unclear at present.

#### 3.1.2. Streptomonomicin

The 21-amino acid lasso peptide, streptomonomicin, belongs to the class II group having no disulfide bonds and is produced by *Streptomonospora alba*. Minimum inhibitory concentrations (MIC) of streptomonomicin range between 2 and 4 µM against *Bacillus anthracis*. The target of streptomonomicin is currently unknown, however resistant mutants were generated against it in *B. anthracis.* These mapped to the response regulator, WalR, suggesting that streptomonomicin might also target peptidoglycan biosynthesis. The same mutation in the DNA-binding domain was found in both streptomonomicin-resistant mutants and siamycin-I resistant mutants (P216S) as well as in the receiver domain (D83Y), which could indicate a similar resistance mechanism for these two peptides [40]. It is interesting to note that streptomonomicin’s core sequence is more similar to that of class I lasso peptides with roughly 50% of its residues being hydrophobic in nature, like that of siamycin-I. It is reasonable then for streptomonomicin to also inhibit lipid II; further mechanistic studies are needed to determine the target and its mode of action

#### 3.1.3. Lariatin A/B

Lariatin A and B are antimicrobials that specifically inhibit *Mycobacterial* growth at 3.13 and 6.25 µg/mL, respectively, in agar diffusion disks. Lariatin A can inhibit *M. tuberculosis* at lower concentrations of 0.39 µg/mL in a microboth dilution assay. The two peptides have no antimicrobial activity against other Gram-positive or Gram-negative pathogens and are therefore hypothesized to interact with the *Mycobacterium* genus’ unique cell envelope. Iwatsuki et al. [54] speculate that given the unique architecture to mycobacterial cell wall and lariatin’s similar characteristics to ethambutol and isoniazid, it is plausible the molecular target of lariatin A/B lies in the cell wall biosynthetic machinery. To date this hypothesis remains untested [54]. Recent work has shed light on which amino acids in lariatin A are important for anti-mycobacterial activity: Tyr-6 (Phe and Trp can be substituted), Gly-11, and Asn-14 were determined to be essential [69].

### 3.2. Inhibitors of RNA Synthesis

More than half of the currently known lasso peptides inhibit bacterial RNA polymerase (RNAP). The most well studied and model lasso peptide, is microcin J25, the first peptide inhibitor of *E. coli* RNAP. Capistruin, acinetodin, klebsidin, and citrocin were later determined to also inhibit bacterial RNAP.

#### 3.2.1. Microcin J25

Microcin J25 is a 21-residue peptide antibiotic produced by *E. coli* harboring a plasmid-borne antibiotic synthesis and export cassette. This compound exhibits potent antimicrobial activity at concentrations as low as 0.01 µg/mL against a range of Gram-negative bacteria [19]. Previously, the target of microcin J25 was thought to be related to *tonB*, *fhuA*, *exbB*, and *sbmA*, all cell envelope proteins required for cellular uptake [70,71]. The complementation of each of the cellular uptake genes could restore microcin J25 sensitivity. However, it was later discovered that microcin J25 targets RNAP. Indeed, a single nucleotide polymorphism was identified in the RNAP ß’ subunit gene, *rpoC*, of a microcin J25-resistant mutant and complementation with an overexpression system of the outer membrane receptor, FhuA, failed to restore microcin J25 activity [72].

To delineate the important residues for antibacterial activity, a mutational analysis was performed and showed that (1) Tyr-9 is critical for inhibition of RNAP and (2) Gly-4, Pro-7, Phe-10, Phe-19, and Tyr-20 are all required for permeation into the bacterial cell and thus, bacterial inhibition [73]. Mechanistic studies were followed by Mukhopadhyay et al. [58] and concluded that microcin J25 inhibits transcript elongation by binding inside RNAP at the secondary channel, also known as the NTP uptake channel. Microcin J25’s ability to block NTPs from entering the RNAP active center gives a partial competitive inhibition to NTPs, which implies that the lasso peptide and NTPs binding sites are distinct, in part or in whole [58].

The recent structure of microcin J25 bound to RNAP indicated that it binds in a position that impairs proper folding of the so-called “trigger loop”, which serves to catalyze the addition of nucleotides on the growing transcript [74]. This compromised folding along with limiting NTP entry and blocks the polymerization of the RNA [74,75] (Figure 6).

#### 3.2.2. Capistruin

As a class II lasso peptide, capistruin is a 19-residue peptide with an antibacterial spectrum parallel to microcin J25. Its MIC ranges from 12 to 50 µM against Gram-negative bacteria. Structurally, capistruin and microcin J25 are very similar so it was thought that capistruin also targeted RNAP. Studies showed that microcin J25-resistant mutants conferred cross-resistance to capistruin both in vitro and in vivo, and this was considered to be conclusive evidence of a shared target [76].

Capistruin also binds the NTP uptake channel and blocks folding of the trigger-loop. However, a co-crystal structure of capistruin-bound RNAP showed that while capistruin’s binding site overlaps that of microcin J25, it is not identical. Consequently, capistruin does not interfere with NTP binding and binds further away from the active catalytic center in RNAP, whereas microcin J25 binds closer to the active center. This is in accordance with capistruin being a non-competitive inhibitor as NTPs can simultaneously bind without much effect on RNAP [74].

#### 3.2.3. Acinetodin and Klebsidin

Acinetodin and klebsidin are lasso peptides that were identified by genome mining human isolates of *Acinetobactera gyllenbergii* and *Klebsiella pneumoniae*, respectively. Acinetodin is an 18-amino acid peptide with no activity against either Gram-positive or Gram-negative bacteria whereas klebsidin is 19 amino acids in length with bioactivity restricted to related species of *Klebsiella* [42].

Since the secretion of some lasso peptides is dependent on a cognate transporter system, the authors went on to heterologously produce both lasso peptides in *E. coli* without the transporter. The biosynthetic cluster was cloned under an inducible promoter, however the ATP-binding cassette transporters, *kleD* and *aciD*, were disrupted by a premature excision. Expression of each disrupted transporter led to intracellular accumulation of the lasso peptides and ceased cellular growth. Therefore, both acinetodin and klebsidin are biologically active lasso peptides when taken up and retained in the cell. To circumvent the lack of bioactivity initially observed, an outer membrane transport protein homologous to FhuA from *K. pneumoniae*, was co-expressed in *E. coli*. This strain was sensitive to klebsidin, but not acinetodin, as expected by the species-specific barrier for this peptide [42].

The similarities in both acinetodin and klebsidin to microcin J25 and capistruin suggested that RNAP is the target of these novel lasso peptides. Indeed, Metelev et al. [42] showed that both can inhibit the elongation step of RNAP in vitro. Wildtype RNAP and microcin J25-resistant RNAP were also assayed with klebsidin and acinetodin. Klebsidin was able to inhibit wildtype RNAP transcription similar to microcin J25 but could not inhibit the resistant RNAP which is mutated at the secondary uptake channel. Similarly, acinetodin inhibited wildtype RNAP but to a lesser extent and was ineffective against the resistant RNAP. Therefore, two more lasso peptides now targeted RNAP at a similar binding site to microcin J25 and capistruin, either partially or in full.

#### 3.2.4. Citrocin

The most recent lasso peptide shown to target RNAP is citrocin. Found from a *Citrobacter* isolate, citrocin is a 19-amino acid peptide with weak inhibitory activity against various *E. coli* strains on disk diffusion assays (16–100 µM) compared to microcin J25. It has been suggested that citrocin differs from microcin J25 in its cellular uptake, mode of action or both. Citrocin-resistant mutants in *E. coli* were obtained and mutations were mapped to *sbmA*, an inner membrane transport protein required for uptake of antimicrobial peptides. A null mutation in *sbmA* confers resistance to citrocin, thus corroborating the hypothesis that citrocin requires an efficient cellular uptake system to exert antibacterial activity. Mutations in *sbmA* were also identified when studying the target of microcin J25 which suggested that the cellular target of citrocin could be RNAP. Interestingly, 1 µM of citrocin can inhibit RNAP in vitro to the same extent as 100 µM of microcin J25, again supporting the hypothesis that uptake of citrocin is a limiting factor [43]. Further structural and mechanistic studies will shed more light on the diversity of class II lasso peptides and their common target, yet different modes of action, against RNAP.

All of the lasso peptides currently known to inhibit RNAP belong to class II, and therefore lack disulfide bridges. Among the five lasso peptides discussed, structurally they each possess a glycine at the 4th position in the core peptide sequence. Additionally, a tyrosine, or phenylalanine in the case of capistruin, is encoded as the last amino acid before the C-terminus (Figure 4A). As mentioned previously, Gly-4 and Tyr-20 are essential for antimicrobial activity in microcin J25. It is interesting to note that class II lasso peptides may have evolved preferentially as RNAP inhibitors whereas the other classes have not.

The varying binding sites on RNAP and modes of action as antibacterials underscore the experimental utility of these molecules and suggest targetable regions for drug discovery against this macromolecular machine.

### 3.3. Inhibitors of the ClpC_1_P_1_P_2_ Protease

The last known target of lasso peptide antibiotics is the ATP-dependent protease complex, ClpC_1_P_1_P_2_, of *M. tuberculosis*. Lassomycin is a 16-residue peptide discovered from a screen of extracts from soil-dwelling bacteria against *M. tuberculosis* followed by a counter-screen for lack of activity against *S. aureus*. MICs were determined to range from 0.8 to 3 µg/mL against various *M. tuberculosis* strains. Resistant mutants of *M. tuberculosis* were obtained and Gavrish et al. [35] observed mutations in the *clpC*1 gene, which encodes the ClpC_1_ subunit. ClpC_1_ is a hexameric ATPase and plays a role in protein degradation, working as a complex with ClpP_1_P_2_ protease.

Enzymatic studies further delineated lassomycin’s mechanism of action. Together, ClpC_1_’s ATP hydrolysis translocates proteins through its channel towards the protease complex, ClpP_1_P_2_. In the presence of lassomycin, ATP hydrolysis was shown to be 7–10-fold higher than in the absence of lassomycin. What was most peculiar was that protein degradation was not accelerated in coordination with the ATPase activity. Instead, proteolysis was completely inhibited in the presence of lassomycin which suggested a new mode of action against bacteria (Figure 6). The binding site of lassomycin was also determined to be distinct from the binding site of proteins the protease recognizes. This led to the hypothesis that lassomycin inhibits the translocation of protein substrates into the proteolytic complex [35].

## 4. Conclusions

Lasso peptides are a promising class of antibiotics given their unique topology and novel mechanisms of action against bacteria. They are also produced by a wide range of bacteria, and the fact that there are so many unrecognized lasso biosynthetic genes in genera such as *Streptomyces* which have been mined for decades, indicates that any estimate of total diversity would be premature at this time.

It also appears that they have diverse and important targets. For example, even with only a small number characterized, the lasso peptides have extended antibiotic target diversity to include the ClpP protease. Further exploration will surely expand this spectrum further and could well provide new therapeutics for circumventing antimicrobial resistance.

Finally, the proteogenic nature of these molecules means that the lasso peptides are amenable to engineering, perhaps more so than any other antibiotics. Currently, work on re-engineering the biosynthesis of lasso peptides and their core sequences is directed at understanding the importance of each residue in bioactivity and biosynthesis. Further investigation into these compounds has the potential to address issues of yield and could drive diversity even further still. This is clearly a rich and promising source of much needed novel bioactive material.

## Figures and Tables

**Figure 1 antibiotics-08-00117-f001:**
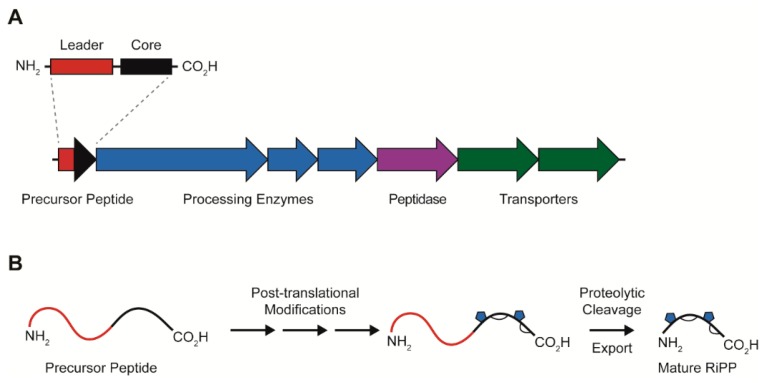
Biosynthesis of ribosomally-synthesized and post-translationally modified peptides (RiPPs). (**A**) General overview of RiPP biosynthetic gene clusters. (**B**) Processing of the precursor peptide into the mature RiPP.

**Figure 2 antibiotics-08-00117-f002:**
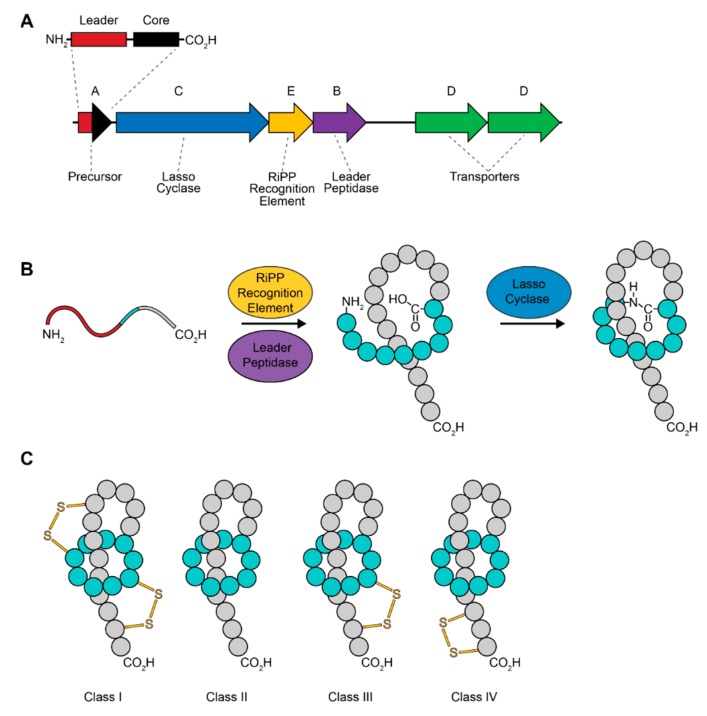
Biosynthesis of lasso peptides. (**A**) General overview of lasso peptide biosynthetic gene clusters including the lasso cyclase, RiPP recognition element (RRE), leader peptidase, and the precursor peptide. (**B**) Processing of the precursor peptide into the mature lasso peptide. (**C**) Comparison of different classes of lasso peptides.

**Figure 3 antibiotics-08-00117-f003:**
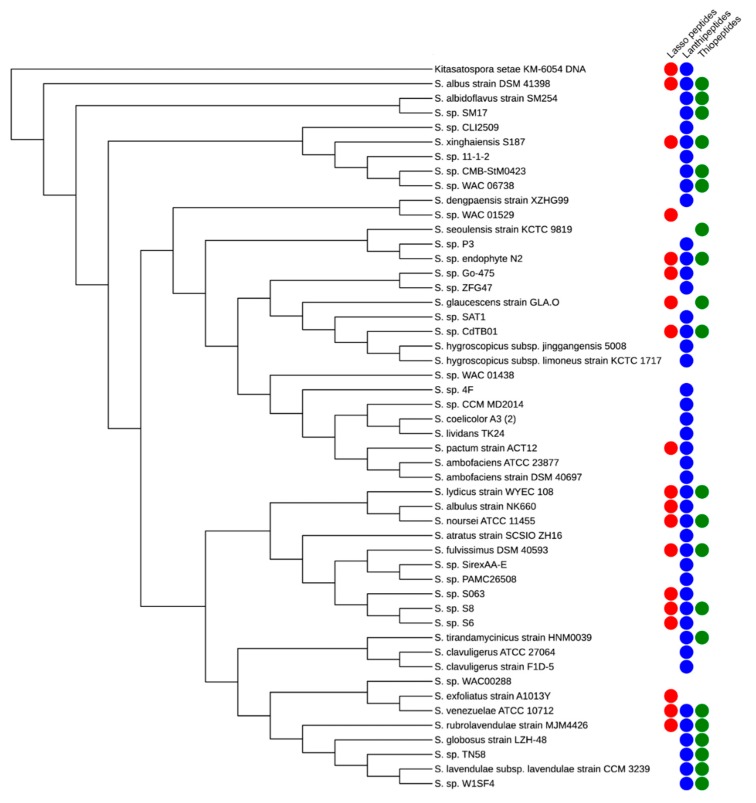
RiPP biosynthetic clusters in *Streptomyces* species. Cladogram was constructed using the sequences of six housekeeping genes (*atpD*, *recA*, *rpoB*, *gyrA*, *trpB*, and 16S rRNA) from randomly-selected *Streptomyces.* Colored circles represent at least one RiPP biosynthetic gene cluster present in the genome. Red circles denote lasso peptide clusters; blue denotes lanthipeptides; green denotes thiopeptides.

**Figure 4 antibiotics-08-00117-f004:**
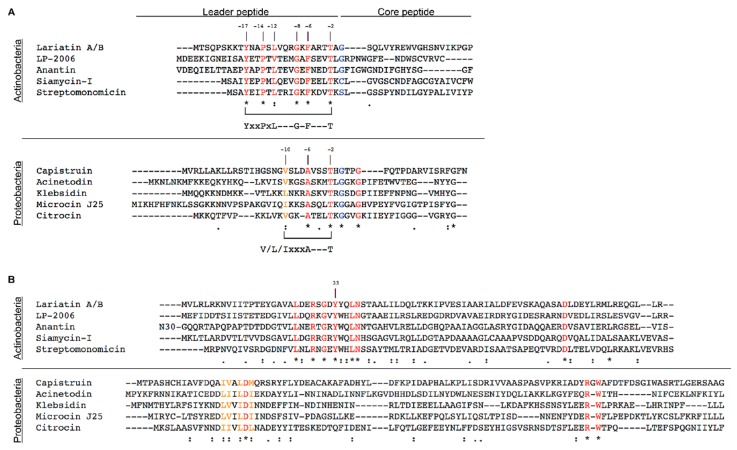
Lasso peptides produced by Actinobacteria or Proteobacteria are different. (**A**) Sequence alignment of the precursor peptides. Lasso peptides produced by Actinobacteria and Proteobacteria differ in their leader peptide. A YxxPxL motif is found in all lasso peptides produced by Actinobacteria whereas those produced by Proteobacteria have a conserved I/V/LxxxA motif. A threonine found two positions upstream of the start of the core peptide is highly conserved in all lasso peptides. (**B**) Sequence alignment of the B1 proteins in actinobacteria and the B proteins in Proteobacteria. No similarities are seen between the two different phyla that produce lasso peptides. N30 represents 30 amino acids not shown in the sequence alignment. Highly conserved residues are highlighted in red; similar residues are highlighted in orange; the start of the core peptide is highlighted in blue. Sequences of each precursor peptide and their respective B protein were accessed through GenBank: lariatin A/B [38], BAL72548.1; LP2006 [24], AFR05970.1; anantin [24], WP_107057732.1; siamycin-I [39], unpublished data; streptomonomicin [40], KIH99640.1; capistruin [41], AIP24900.1; acinetodin [42], EPF88080.1; klebsidin [42], WP_023288230; microcin J25 [32], AGC14196.1; citrocin [43], OQM39068.1.

**Figure 5 antibiotics-08-00117-f005:**
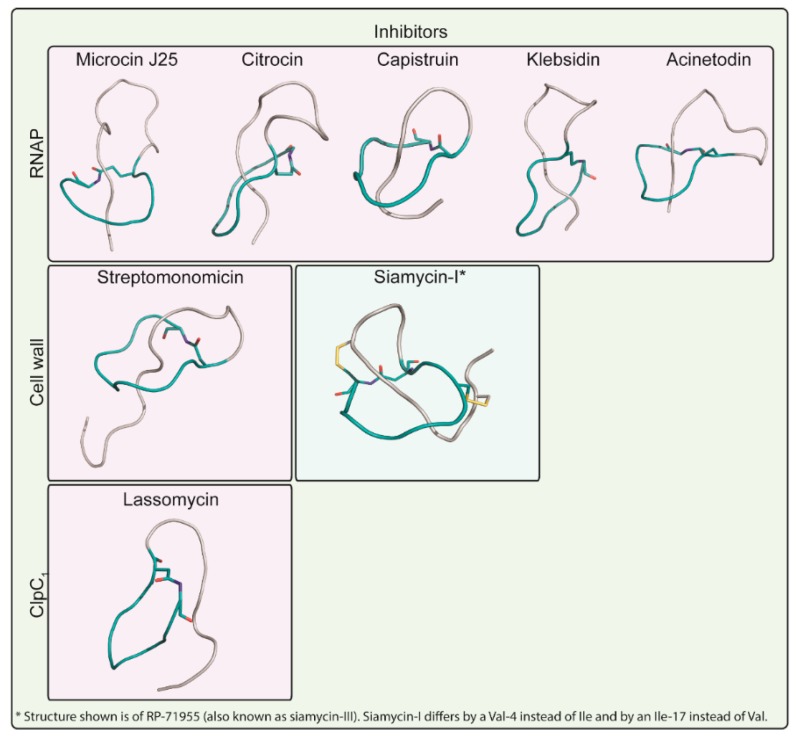
Structures of lasso peptides with a known molecular target. Microcin J25, citrocin, capistruin, klebsidin, and acinetodin are all lasso peptides that target RNA polymerase (RNAP). Streptomonomicin and siamycin-I are known lasso peptide inhibitors of the bacterial cell wall. Lassomycin is the only lasso peptide known to inhibit the ClpC1 unit of the ClpC1P1P2 protease complex. All the lasso peptides in purple boxes are class II lasso peptides with no disulfide bridges, whereas only siamycin-I, in a blue box, is a class I lasso peptide with two disulfide bridges, which are shown in yellow. The side chains of the amino acids involved in macrolactam formation are colored according to their elemental composition.

**Figure 6 antibiotics-08-00117-f006:**
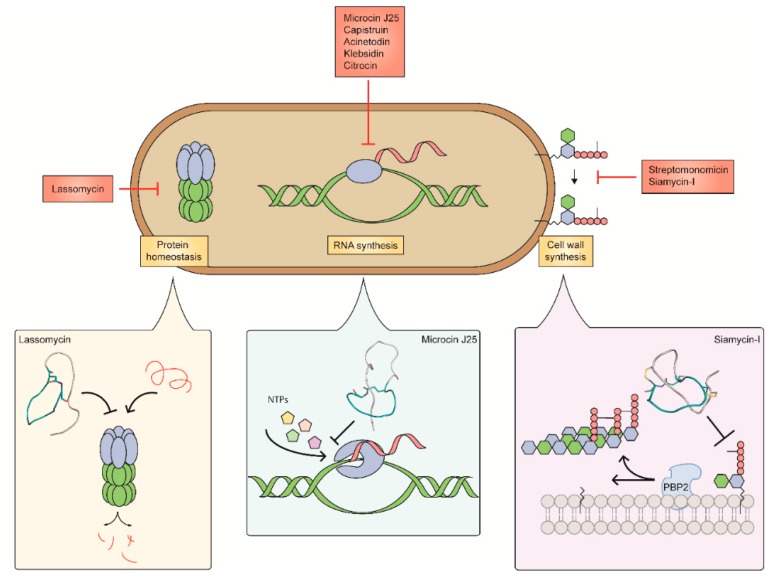
Targets and mode of action of lasso peptides. The known antibacterial targets of lasso peptides include inhibiting protein homeostasis, RNA synthesis, and cell wall biosynthesis. The mechanism of action of lasso peptides is shown for lassomycin, the only protein homeostasis inhibitor, microcin J25, the model lasso peptide and first to be shown as an inhibitor of RNAP, and siamycin-I, the first lasso peptide to target lipid II in peptidoglycan.

**Table 1 antibiotics-08-00117-t001:** Antibiotic activity of lasso peptides.

Lasso Peptide	Producer Strain	Antibiotic Activity		No Antibiotic Activity	
**CLASS I**					
Siamycin-I ^¥^	*Streptomyces* sp.	*B. subtilis* ^+^ *S. epidermis* ^+^ *S. saprophyticus* ^+^	*S. aureus* ^+^ *E. faecalis* ^+^	*A. baumannii* ^−^ *B. cepacian* ^−^ *P. aeruginosa* ^−^	*E. coli* ^−^ *K. pneumoniae* ^−^
Aborycin ^‡^	*Streptomyces griseoflavus* TU4072 *Streptomyces* sp. 9440	*B. brevis* ^+^ *E. fallinarum* ^+^ *B. subtilis* ^+^ *S. viridochromeogenes* ^+^	*E. faecalis* ^+^ *S. aureus* ^+^ *B. thuringiensis* ^+^ *P. saccharophilia* ^−^	*A. baumannii* ^−^ *M. luteus* ^+^ *C. perfringens* ^+^ *S. typhirium* ^−^	*E. coli* ^−^ *V. alginolyticus* ^−^ *K. pneumoniae* ^−^
**CLASS II**					
Acinetodin ^¥^	*Acinetobacter gyllenbergii* (clinical gut isolate)	*E. coli* * ^−^		*A. baumannii* ^−^ *B. subtilis* ^+^ *S. aureus* ^+^	*K. pneumoniae* ^−^ *P. aeruginosa* ^−^
Anantin B2 ^‡^	*Streptomyces coerulescens*	*B. subtilis* ^+^ *E. coli* ^−^		*A. baumannii* ^−^ *B. anthracis* ^+^ *M. smegatis* ^+^ *N. meningitidis* ^−^ *L. monocyte-genes* ^+^	*E. coli* ^−^ *P. aeruginosa* ^−^ *K. pneumoniae* ^¯^ *S. aureus* ^+^ *E. faecium* ^+^
Astexin-1 ^¥^	*Asticcacaulis excentricus* CB48	*C. crescentus* ^−^		*B. thailandensis* ^−^ *V. harveyi* ^−^ *E. coli* ^−^	*S. newport* ^−^ *V. fischeri* ^−^
Capistruin ^¥^	*Burkholderia thailandensis* E264	*Burkholderia* sp. ^−^ *E. coli* ^−^ *P. aeruginosa* ^−^		*A. viridans*^+^ *Pseudomonas* sp. ^−^ *B. megaterium* ^+^	*K. pneumoniae* ^−^ *S. enterica* ^¯^ *S. aureus* ^+^
Citrocin ^¥^	*Citrobacter braakii* ATCC 51113	*Citrobacter* sp. ^−^ *E. coli* ^¯^		*P. aeruginosa* ^−^ *S. marcescens* ^−^	
Klebsidin ^¥^	*Klebsiella pneumoniae* (clinical gut isolate)	*E. coli* * ^−^ *K. pneumoniae* ^−^		*A. baumannii* ^−^ *P. aeruginosa* ^−^	*B. subtilis* ^+^ *S. aureus* ^+^
Lariatin A/B ^¥^	*Rhodoccocus jostii* K01-B0171	*M. smegatis* ^+^ *M. tuberculosis* ^‡ +^		*P. aeruginosa* ^−^ *B. subtilis* ^+^ *S. aureus* ^+^	*E. coli* ^−^ *X. campestris* ^−^ *M. luteus* ^+^
Lasssomycin ^‡^	*Lentzea kentuckyensis*	*M. tuberculosis*^+^ *Mycobacterium* spp. ^+^		*B. anthracis*^+^ *K. pneumoniae*^−^ *C. difficile*^+^ *Lactobacillus* sp. ^+^	*E. faecalis* ^+^ *S. mutans* ^+^ *E. coli* ^−^ *S. aureus* ^+^
Microcin J25 ^¥^	*Escherichia coli*	*E. coli* ^−^ *S. flexneri* ^−^	*S. newport* ^−^	*P. mendocina* *L. acidophilus* ^+^ *B. subtilis* ^+^ *S. enterica* ^−^	*K. pneumoniae* ^−^ *S. typhirium* ^−^
Propeptin 1/2 ^¥^	*Microbispora* sp. SNA-115	*M. phlei* ^+^ *X. orzyae* ^−^	*P. aeruginosa* ^−^	N/A	
Streptomono -micin ^‡^	*Streptomonospora alba* YIM	*B. subtilis* ^+^ *L. monocyte* ^−^ *genes* ^+^	*B. anthracis* ^+^ *B. cereus* ^+^ *S. aureus* ^+^	*E. coli* ^−^	*P. aeruginosa* ^−^
**CLASS IV**					
LP2006 ^‡^	*Nocardiopsis alba*	*B. anthracis* ^+^ *B. subtilis* ^+^	*E. faecalis* ^+^ *M. smegatis* ^+^	*A. baumannii* *N. meningitidis* ^−^ *E. coli* ^−^ *P. aeruginosa* ^−^	*K. pneumoniae* ^−^ *S. aureus* ^+^ *L. monocyte* ^−^ *genes* ^+^

^‡^ Antibiotic activity was measured by microtiter broth dilution. ^¥^ Antibiotic activity was measured by paper disk diffusion assay. * Antibiotic activity was reported after heterologously expressing the lasso peptide in *E. coli* without its respective exporter/transporter, leading to intracellular accumulation. ^+/−^ Gram-positive and Gram-negative bacteria, respectively.

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
