# Peer review of "Put a Bow on It: Knotted Antibiotics Take Center Stage"

_antibiotics, 2019, doi:10.3390/antibiotics8030117_

Round 1

Reviewer 1 Report

In this review “Put a bow on it: knotted antibiotics take centre stage” authors described biosynthesis, structures, antibacterial activities, and molecular targets of lasso peptides. The review is well written and will be suitable for publication after following minor changes.

1. Line 118-120, a figure/scheme describing lasso peptide cyclization will be helpful for general readers.

2. Considering the broad antibacterial activity and large structures, lasso peptides might not obey Lipinski’s rule. Therefore, a discussion on their structural features responsible for antibacterial activity and Lipinski rule will help the readers.

3. Table 1, different colors for gram-positive and negative pathogen will help.

4.  Wherever possible, the authors should provide the MIC/IC50 of lasso peptides for the observed antibacterial activity.

Author Response

Minor Revisions

Line 118-120, a figure/scheme describing lasso peptide cyclization will be helpful for general readers.

Comment/Rebuttal:

A reference to Figure 2b has been added to the section where lasso peptide biosynthesis is discussed. This should aid the general reader with visualizing the process of lasso peptide biosynthesis as it depicts the processes of cyclization and proteolysis and identifies the enzymes involved. To date, lasso peptide biosynthesis is not understood well enough to depict biosynthesis at the mechanistic level.

Considering the broad antibacterial activity and large structures, lasso peptides might not obey Lipinski’s rule. Therefore, a discussion on their structural features responsible for antibacterial activity and Lipinski rule will help the readers.

Comment/Rebuttal:

A short discussion in regard to Lipinski’s rules have been added to the text. Please see Lines 42-54.

The majority of studies have looked at the amino acids within the entire precursor peptide to determine which residues are important for maturation of the lasso peptide to further understand the biosynthesis of lasso peptides. A few examples are listed here:

Knappe, T.A., Linne, U., Robbel, L. & Marahiel, M.A. Insights into the Biosynthesis and Stability of the Lasso Peptide Capistruin. Cell Chem. Bio. 2009, 16, 1290-1298 Zimmermann, M.; Hegemann, J. D.; Xie, X.; Marahiel, M. A. The Astexin-1 Lasso Peptides: Biosynthesis, Stability, and Structural Studies. Cell Chem. Biol. 2013, 20, 558–569. Yan, K.-P.; Li, Y.; Zirah, S.; Goulard, C.; Knappe, T. A.; Marahiel, M. A.; Rebuffat, S. Dissecting the Maturation Steps of the Lasso Peptide Microcin J25 in Vitro. ChemBioChem 2012, 13, 1046–1052.

From the gathered literature, only 2 of the lasso peptides discussed in detail (lariatin A and microcin J25) have undergone mutational analyses in their core peptide to determine which amino acids are required for anti-mycobacterial and antibacterial activity, respectively. These two examples have been added into the text: L312-314 and L330-333.

There is currently no general consensus of which amino acids are critical for antibacterial activity given that there are a variety of targets currently and their interactions have yet to be looked at more in depth. The only molecules that have been crystallized bound to their target are those that inhibit RNAP and this was only seen in microcin J25 and capistruin [1]. There have been attempts at doing mutational analyses in capistruin, however the production yield has been a difficult step to overcome [2]

Braffman, N. R.; Piscotta, F. J.; Hauver, J.; Campbell, E. A.; Link, A. J.; Darst, S. A. Structural Mechanism of Transcription Inhibition by Lasso Peptides Microcin J25 and Capistruin. Natl. Acad. Sci. 2019, 116, 1273–1278. Knappe, T.A., Linne, U., Robbel, L. & Marahiel, M.A. Insights into the Biosynthesis and Stability of the Lasso Peptide Capistruin. Chemistry & Biology. 2009 16, 1290-1298

A section on the structural features is inferred from the 5 known lasso peptides that target RNAP found in Lines 392-397.

Table 1, different colors for gram-positive and negative pathogen will help.

Comment/Rebuttal:

An indication has been added to Table 1 to differentiate between Gram-positive and Gram-negative bacteria.

Wherever possible, the authors should provide the MIC/IC50 of lasso peptides for the observed antibacterial activity.

Comment/Rebuttal:

MICs have been added into the text for each of the lasso peptide as discussed: Lines 250-251 (siamycin-I), Lines 293-294 (streptomonomicin), Lines 305-307 (lariatins), Lines 322-323 (microcin J25), Line 344 (capistruin), Line 380 (citrocin) and Line 405-406 (lassomycin).

Reviewer 2 Report

I found the article very informative. I had little or no knowledge of the lasso peptides and it is fascinating that so many peptides with this peculiar structure should exist while so little is known about them. The article shows there is a lot of interesting biochemistry here, not to mention the possibility of important biomedical applications. I don’t have any strong criticisms. I will make a few suggestions for improvement, but it’s nit-picking really.

The article is a little hard to read in places. It occasionally assumes more background knowledge than a general reader will have. For example, a few times I found myself needing to look up terms that could have been explained in the text, even if only as asides. For example, lines 97-100 speak of lanthapeptide, thiopeptide, azol(in)e-containing, and linaridin clusters. The uninitiated reader will have little idea what these are. Of course, the main point is apparently only to show that streptomycetes make a host of RiPPs, but it would be helpful if the reader could be reminded how they differ structurally from lasso peptides. I know this is not intended to be a review of RiPPs in general, but a little more RiPP context would be helpful for the non-expert.

In places the text has a meandering quality that makes it hard to distinguish essential points from less important ones. For example, the descriptions of the activities of some of the peptides wandered through the history of their characterization, only arriving at the main conclusions at the end of the section. This is all interesting, but an up-front mention of the section’s conclusions would give the reader the essential message right away and then experimental context could come after. Another example: Lines 226-230 tell us that siamycin-I has a variety of apparently unrelated activities (HIV receptor inhibition, etc.) before going on to a more extensive account of how it has a much better characterized inhibitory activity in peptidoglycan synthesis. This organization puts the reader off the track at the outset.  Lines 226-230 perhaps could be put at the end of the section and these points mentioned only as other activities whose full significance is unknown at present.

A couple of very minor things:

163-164 – “Contrary, Proteobacteria encode a leader peptide….” Is a peculiar construction (even if understandable).

300 - Something is wrong in this line - probably just an extra “with” I think.

Author Response

I found the article very informative. I had little or no knowledge of the lasso peptides and it is fascinating that so many peptides with this peculiar structure should exist while so little is known about them. The article shows there is a lot of interesting biochemistry here, not to mention the possibility of important biomedical applications. I don’t have any strong criticisms. I will make a few suggestions for improvement, but it’s nit-picking really.

The article is a little hard to read in places. It occasionally assumes more background knowledge than a general reader will have. For example, a few times I found myself needing to look up terms that could have been explained in the text, even if only as asides. For example, lines 97-100 speak of lanthapeptide, thiopeptide, azol(in)e-containing, and linaridin clusters. The uninitiated reader will have little idea what these are. Of course, the main point is apparently only to show that streptomycetes make a host of RiPPs, but it would be helpful if the reader could be reminded how they differ structurally from lasso peptides. I know this is not intended to be a review of RiPPs in general, but a little more RiPP context would be helpful for the non-expert.

Comment/Rebuttal:

To simplify the figure and the discussion about RiPPs, the linear azol(in)e-containing peptide, and linaridin clusters have been removed from Figure 3. An additional explanation of lanthipeptides and thiopeptides has been added to the text which we believe clarifies these other classes of RiPPs for readers who are unfamiliar with them. Please see lines 108-111 and Figure 3 for the changes.

In places the text has a meandering quality that makes it hard to distinguish essential points from less important ones. For example, the descriptions of the activities of some of the peptides wandered through the history of their characterization, only arriving at the main conclusions at the end of the section. This is all interesting, but an up-front mention of the section’s conclusions would give the reader the essential message right away and then experimental context could come after. Another example: Lines 226-230 tell us that siamycin-I has a variety of apparently unrelated activities (HIV receptor inhibition, etc.) before going on to a more extensive account of how it has a much better characterized inhibitory activity in peptidoglycan synthesis. This organization puts the reader off the track at the outset.  Lines 226-230 perhaps could be put at the end of the section and these points mentioned only as other activities whose full significance is unknown at present.

Comment/Rebuttal:

This has been addressed in the following places: Lines 213-215, Lines 248-252 and Lines 285-287.

Minor Revisions

163-164 – “Contrary, Proteobacteria encode a leader peptide….” Is a peculiar construction (even if understandable).

Comment/Rebuttal:

“Contrary” has been changed to “By contrast”, which we believe clarifies the sentence.

300 - Something is wrong in this line - probably just an extra “with” I think.

Comment/Rebuttal:

“with” removed from the sentence. Please see line 328.

Reviewer 3 Report

1.     General comments

In this review, current knowledge on lasso peptide biosynthesis as well as their antibiotic activity, molecular targets and mechanisms of action was well discussed.

The review represents a great advance in the understanding the significance of lasso peptide as novel bioactive material.

2.     Major revision

1)    Figure 4.

It is strongly recommended to revise the figure so that the reader can easily understand.

A) It is essential to show the reference No. or accession No. of amino acid sequence in 10 lasso peptides (Lariatin A/B ~ Citrocin).

B) It is strongly recommended to draw the regions of YxxPxL motif and I/V/LxxxA motif in Figure 4a.

C-1) It is strongly recommended to draw the position of threonine (T) found 2 positions upstream of the start of the core peptide in Figure 4, similarly to Figure 3 of Ref. 35).

35) Sumida, T.; Dubiley, S.; Wilcox, B.; Severinov, K.; Tagami, S. Structural Basis of Leader Peptide Recognition in Lasso Peptide Biosynthesis Pathway. ACS Chem. Biol. 2019.

C-2) For ref. 35), it is essential to revise “ACS Chem. Biol. 2019.” to  “ACS Chem. Biol. 2019, 14, 1619−1627.

D) Line 165-166: It is strongly recommended to draw the position of “Phe-6 in leader peptide and Tyr-33 in B1 protein (line 165-166)” in Figure 4a.

3.     Minor revision

1)    Figure 2: It is recommended to choose (unify) one of three words “Leader Recognition (Figure 2a)”, “RiPP Recognition element (Figure 2b)” and “leader recognition element (RRE) (legend of Figure 2)”.  

2)    Figure 5: It is recommended to explain the side chain of amino acid residue shown as red or yellow color.

3)    Line 231: It is recommended to revise “Daniel-Ivad et al. to “Daniel-Ivad et al. [45]”

4)    Line 341: It is recommended to revise “Metelev et al.” to “Metelev et al. [48]”.

5)    It is essential to check and revise the following references in Line 409, Line 419, Line 457, Line 459, Line 554 and Line 566.

Author Response

Major Revisions

Figure 4. It is strongly recommended to revise the figure so that the reader can easily understand. It is essential to show the reference No. or accession No. of amino acid sequence in 10 lasso peptides (Lariatin A/B ~ Citrocin). It is strongly recommended to draw the regions of YxxPxL motif and I/V/LxxxA motif in Figure 4a. It is strongly recommended to draw the position of threonine (T) found 2 positions upstream of the start of the core peptide in Figure 4, similarly to Figure 3 of Ref. 35). Line 165-166: It is strongly recommended to draw the position of “Phe-6 in leader peptide and Tyr-33 in B1 protein (line 165-166)” in Figure 4a.

Comment/Rebuttal:

Figure 4. has been updated to include the accession numbers and references of the amino acid sequences in the alignment. The YxxPxL motif and I/V/LxxxA motif regions have been highlighted, as have the T-2 and F-6 positions. We believe that these changes have clarified the figure.

Sumida, T.; Dubiley, S.; Wilcox, B.; Severinov, K.; Tagami, S. Structural Basis of Leader Peptide Recognition in Lasso Peptide Biosynthesis Pathway. ACS Chem. Biol. 2019. For ref. 35), it is essential to revise “ACS Chem. Biol. 2019.” to “ACS Chem. Biol. 2019,14, 1619−1627.”

Comment/Rebuttal:

The reference has been updated to include the journal volume and the page numbers. Please see lines 538-539 for such changes.

Minor Revisions

Figure 2: It is recommended to choose (unify) one of three words “Leader Recognition (Figure 2a)”, “RiPP Recognition element (Figure 2b)” and “leader recognition element (RRE) (legend of Figure 2)”.

Comment/Rebuttal:

All mentions of the RiPP recognition element have been changed to “RiPP recognition element”, including in Figure 2 and the caption of Figure 2.

Figure 5: It is recommended to explain the side chain of amino acid residue shown as red or yellow color.

Comment/Rebuttal:

An additional explanation of the colour scheme in Figure 5 has been added to the caption.

Line 231: It is recommended to revise “Daniel-Ivad et al. to “Daniel-Ivad et al. [45]”

Comment/Rebuttal:

A mid-sentence reference has been added. Please see line 248.

Line 341: It is recommended to revise “Metelev et al.” to “Metelev et al. [48]”.

Comment/Rebuttal:

A mid-sentence reference has been added. Please see line 370.

It is essential to check and revise the following references in Line 409, Line 419, Line 457, Line 459, Line 554 and Line 566.

Comment/Rebuttal:

The references listed above have been updated to include any missing information and formatted according to the MDPI reference guidelines.